# Detection and genetic characterisation of *Toxoplasma gondii* circulating in free-range chickens, pigs and seropositive pregnant women in Benue state, Nigeria

**Ifeoma N. Nzelu**[1¤], **Jacob K. P. Kwaga**[2], **Junaidu Kabir**[2], **Idris A. Lawal**[3], **Christy Beazley**[1], **Laura Evans**[1], **Damer P. Blake**[1] *

1 Department of Pathobiology and Population Sciences, Royal Veterinary College, University of London, Hertfordshire, United Kingdom, 2 Department of Veterinary Public Health and Preventive Medicine, Faculty of Veterinary Medicine, Ahmadu Bello University, Zaria, Kaduna State, Nigeria, 3 Department of Veterinary Parasitology and Entomology, Faculty of Veterinary Medicine, Ahmadu Bello University, Zaria, Kaduna State, Nigeria

¤ Current address: Department of Veterinary Public Health and Preventive Medicine, College of Veterinary Medicine, Joseph Sarwuan Tarka University Makurdi, Nigeria
* dblake@rvc.ac.uk

**Data Availability Statement:** The authors confirm that all data underlying the findings are fully available without restriction. All sequences

## Abstract

*Toxoplasma gondii* parasites present strong but geographically varied signatures of population structure. Populations sampled from Europe and North America have commonly been defined by over-representation of a small number of clonal types, in contrast to greater diversity in South America. The occurrence and extent of genetic diversity in African *T. gondii* populations remains understudied, undermining assessments of risk and transmission. The present study was designed to establish the occurrence, genotype and phylogeny of *T. gondii* in meat samples collected from livestock produced for human consumption (free-range chickens, n = 173; pigs, n = 211), comparing with *T. gondii* detected in blood samples collected from seropositive pregnant women (n = 91) in Benue state, Nigeria. The presence of *T. gondii* DNA was determined using a published nested polymerase chain reaction, targeting the 529 bp multicopy gene element. Samples with the highest parasite load (assessed using quantitative PCR) were selected for PCR-restriction fragment length polymorphism (PCR-RFLP) targeting the surface antigen 3 (SAG3), SAG2 (5' and 3'), beta-tubulin (BTUB) and dense granule protein 6 (GRA6) loci, and the apicoplast genome (Apico). *Toxoplasma gondii* DNA was detected in all three of the populations sampled, presenting 30.6, 31.3 and 25.3% occurrence in free-range chickens, pigs and seropositive pregnant women, respectively. Quantitative-PCR indicated low parasite occurrence in most positive samples, limiting some further molecular analyses. PCR-RFLP results suggested that *T. gondii* circulating in the sampled populations presented with a type II genetic background, although all included a hybrid type I/II or II/III haplotype. Concatenation of aligned RFLP amplicon sequences revealed limited diversity with nine haplotypes and little indication of host species-specific or spatially distributed sub-populations. Samples collected from humans shared haplotypes with free-range chickens and/or pigs. Africa remains under-

generated here have been submitted to the European Nucleotide Archive under the project reference PRJEB42973, with accession numbers HG992860-HG992962. Sequences can be accessed via GenBank (https://www.ncbi.nlm.nih.gov/nucleotide/). Sequence alignments are available from the Mendeley data repository under the accession http://dx.doi.org/10.17632/fscy8463cd.2.

**Funding:** This research was funded by the UK government through an award to INN as a Commonwealth PhD Scholar under the reference NGCN-2016-165 (https://cscuk.fcdo.gov.uk/scholarships/). The funders had no role in study design, data collection and analysis, decision to publish, or preparation of the manuscript.

**Competing interests:** The authors have declared that no competing interests exist.

explored for *T. gondii* genetic diversity and this study provides the first detailed definition of haplotypes circulating in human and animal populations in Nigeria.

## Author summary

*Toxoplasma gondii* is a parasite that infects most warm-blooded animals and can be transmitted from animals to humans. Three dominant genetic types have been described from a larger pool of around 16, and it has been suggested that the severity of disease may be influenced by genetic type. Little is known of *T. gondii* in Africa. The burden of disease is unclear, while lack of knowledge around genetic diversity and population structure undermines effective risk assessment and control. We sought to determine if *T. gondii* was prevalent in pigs and poultry produced for human consumption in Nigeria, comparing with genetic types detected in the overlapping human population. Using meat samples from free-range chickens and pigs, and blood samples from seropositive pregnant women in Benue state, Nigeria, we found that *T. gondii* with a type II genetic background were most common with limited genetic diversity. Detection of comparable genetic types in the free-range chicken, pig and human populations indicate an overlapping parasite population and can be used to inform assessments of risk to human health, most notably pregnant women. The information reported here informs on the occurrence and population structure of *T. gondii* in Nigeria, contributing to improved understanding in Africa.

## Introduction

*Toxoplasma gondii* is a zoonotic protozoan parasite with a global distribution. Members of the Felidae, especially domestic cats, serve as the definitive hosts of the parasite, while humans, birds and other warm-blooded animals play a role as intermediate hosts in the transmission cycle [1]. Cats primarily become infected with *T. gondii* by eating tissues of intermediate hosts harbouring cysts, while food animals including chickens and pigs most commonly become infected by ingestion of oocysts from the environment contaminated with cat faeces [2,3]. Eating raw or undercooked meat containing tissue cysts, consuming water or food contaminated with oocysts from cat faeces, or inadvertently ingesting oocysts from the environment are the commonest routes for human infection. Free-range chickens play an important role in the epidemiology of *T. gondii* infection [2]. They are regarded as good indicators for soil contamination with *T. gondii* oocysts, serving as sentinels due to their scavenging feeding habits [4]. Tissues of infected chickens are also considered a common source of infection for cats [4]. Several studies have reported varying seroprevalence of *T. gondii* in chickens and pigs [5–7], as well as detection of *T. gondii* DNA in chicken and pork meat products [8,9], raising concerns of possible risk to humans. Bioassays remain the most sensitive means of detecting *Toxoplasma* cysts in animal tissues [10], but these methods are laborious, time-consuming and are poorly adapted to the screening large numbers of samples [10]. As a result, PCR-based methods have been developed to detect parasite DNA in meat samples as an indication of exposure risk [11,9], with methods such as Magnetic-Capture real-time PCR (MC-RT PCR) and acid pepsin digest (PD-) RT PCR offering best sensitivity [12].

The population structure of *T. gondii* was initially described as highly clonal, exhibiting low genetic diversity with three main lineages, types I, II, and III, based upon studies from North America and Europe [13,14]. However, more recent studies have revealed greater diversity in

*T. gondii* populations sampled from other continents such as South America [15,16,17,18], Africa and Asia [19]. In Africa, the three most globally over-represented clonal lineages are supplemented by haplogroups named Africa 1, 2 and 3 [20,21,22], although remarkably little is known of their occurrence, genetic diversity or population structure. For Nigeria, just one report has described detection and genetic characterisation of *T. gondii*, relying on a single sample recovered from a chicken [20]. Surveys based upon seroprevalence using pigs and chickens indicate a high level of *T. gondii* occurrence [2,3,5,7,23,24,25,26], but population genetic data is scarce. In addition, there is a similar dearth of information regarding the *T. gondii* haplotypes circulating in humans in Nigeria and other African countries. Such a deficit precludes the attribution of infection sources, undermining development of effective strategies for public health. In response, the present study was designed to identify *T. gondii* haplotypes circulating in human and animal populations in Benue State, Nigeria, during a six-month sampling window in 2016, successfully generating new genetic data.

## Methods

### Ethics statement

Ethical approval was obtained from the Ahmadu Bello University Committee on Animal Use and Care (ABUCAUC), Ahmadu Bello University Committee on the Use of Human Subject for Research (ABUCUHSR; reference ABUCUHSR/HSR/2016/001), Benue State Hospital Management Board (HMB/MED/40/VOL.II/211), Federal Medical Center, Makurdi (FMH/ FMC/MED.108/VOL1/X) and Clinical Research Ethical Review Board (CRERB) of the Royal Veterinary College (RVC), London with approval number URN 2016 1614. For human subjects, informed consent was obtained following explanation of the significance of the study and assurance of confidentiality. Participants were requested to sign a document confirming their permission to be included in the study, recognising their right of withdrawal at any time.

### Study area

The study was conducted in Benue State, Nigeria, located in the North central geo-political zone of Nigeria. Its geographic coordinates are 7.3369˚ N and 8.7404˚ E. Benue State shares boundaries with five other states, namely; Nasarawa State to the north, Taraba State to the east, Cross-River State to the south, Enugu State to the south-west and Kogi State to the west, as well as the Republic of Cameroon to the southeast. The state experiences two seasons; a wet season from April to October, and a dry season from November to March. Inhabitants of the State are commonly engaged in farming, fishing and animal husbandry.

### Sampling procedure

A two-stage sampling technique was used for the selection of sampling units from Benue State. Benue state has three senatorial zones. Samples were collected within each zone from selected Local Government Areas (LGA). For free-range chickens, sampling units were obtained from the identified major live bird market from each selected LGA, and identified slaughter slabs for pigs. For human subjects, samples were collected from the State General Hospital in each selected LGA, and from the Federal Medical Center (FMC) in the state capital, Makurdi. Selected hospitals in the study area were approached for their consent for inclusion in the study and ethical clearance was obtained as stated above. Local government areas were selected based on simple balloting. They included Ukum, Kastina-Ala, Makurdi, Gboko, Otukpo and Okpokwu for free-range chickens; Ukum, Kastina-Ala, Konshisha, Makurdi, Gboko and

Gwer-East for pigs, and Kasitina-Ala, Makurdi and Otukpo for pregnant women. Study subjects were selected by convenience during sampling visits.

## Sample collection

Samples for the current study were collected from June to December 2016. For animals reared for human consumption ~25 g thigh muscle was collected from each sampled chicken or pig using scissors and a knife, disinfected using alcohol between samples. Each tissue sample was placed into a clean, labelled polyethylene bag. All samples were kept in a cool box containing wet ice and transported to the Ahmadu Ali Center for Public Health and Comparative Medicine laboratory, Makurdi, for processing. Tissue samples were stored at -20 °C prior to processing.

For human subjects, ~3 ml of blood was aseptically collected from each participant via the median cubital vein by a trained medical laboratory scientist. Approximately 2.5 ml whole blood was dispensed into a 5 ml plain vacutainer (for serology; as described previously [27]), with the remaining 0.5 ml placed in an EDTA sample bottle. Samples were placed in a cool box containing ice and transported to the Ahmadu Ali Center for Public Health and Comparative Medicine laboratory, Makurdi, for processing. Using a micropipette with disposable tips, 125 μl of each EDTA-stored blood sample was transferred onto the sample area on a Whatman FTA card (GE Healthcare UK Limited, United Kingdom). The cards were allowed to dry, and stored at room temperature under ambient conditions for up to one year prior to processing for DNA extraction.

## DNA extraction

Total genomic DNA was extracted from frozen tissue samples using the Qiagen DNeasy blood and tissue kit (Qiagen, USA). Briefly, each frozen sample was thawed at room temperature, after which ~1 g was excised, weighed and placed into a labelled, sterile 30 ml universal sample bottle (StarLab, UK) containing 2 ml Qiagen buffer ATL for homogenisation using a Qiagen tissue ruptor. After homogenisation, an aliquot of each sample was processed as described by the manufacturer. Eluates were stored at -20˚C until required as template for PCR.

For human blood, DNA was extracted from samples preserved on Whatman FTA cards using 5% (w/v) Chelex 100 resin (Bio-Rad Laboratories, France) prepared in molecular grade water (Thermo Fisher Scientific, UK) [28]. Samples collected from women found to be sero-positive for *T. gondii* (IgM and/or IgG, or found to be inconclusive [27]) were prioritised to improve parasite detection. Specifically, 83 samples had previously achieved a definitive result for IgM (14 positive, 69 negative), with 9 found to be inconclusive. Similarly, 89 samples achieved a a definitive result for IgG (73 reactive, 16 non-reactive), with 2 found to be inconclusive. Briefly, a Harris Uni–Core disposable punch (2.0) was used to remove ten dried blood discs from the centre of each FTA card, representing ~20% of each blood sample. Discs were soaked in 1× Tris- ethylenediamine tetraacetic acid (EDTA) (Ultrapure UK) (TE) buffer solution and incubated at room temperature for 10 mins. After soaking, the TE buffer was removed and the soaking procedure repeated. Washed discs were centrifuged at 12,000 g for 3 min, the supernatant was removed and replaced with 200 μl Chelex 100 suspension. Samples were then incubated at 56˚C for 20 min, vortex mixed for 15 sec, incubated at 100˚C for 8 min, and vortex mixed again. Samples were centrifuged at 12,000 g for 3 min, then the supernatant was recovered and stored at -20˚C prior to use as template for PCR. Discs removed from empty parts of FTA cards and processed in parallel were used to provide negative control for sample storage, processing and subsequent molecular analysis.

All DNA samples were tested using a Nanodrop (Denovix, DS-11 spectrophotometer) to determine the concentration (ng/μl) and quality of DNA at an absorbance ratio of 260/280. Samples with an absorbance ratio of 1.8 and above were retained for use (DeNovix, 2017).

## PCR detection of *T. gondii* DNA

The presence of *T. gondii* genomic DNA was detected using a nested PCR targeting the 529 bp multicopy gene element (GenBank Accession No. AF146527) as described by Kong *et al.* [29] with modified cycling conditions. Primary amplification was optimised in a 50 μl reaction volume containing 25 μl 2× MyTaq Mix (Bioline), 0.2 μl primers NF1 and NR1 (5'-TGACTCGG GCCCAGCTGCGT-3' and 5'-CTCCTCCCTTCGTCCAAGCCTCC-3', both 100 μM), 4 μl extracted DNA and 20.6 μl Ultrapure molecular grade water (ThermoFisher Scientific, UK). The reaction mixture was heated at 94˚C for 1 min, followed by 35 cycles of 94˚C for 1 min, 58˚C for 1 min and 72˚C for 1 min, with a final extension step of 72˚C for 10 min using a G-Storm thermal cycler. Second round nested PCR amplification was optimised in a 25 μl reaction volume, using 1 μl of the first round PCR product (diluted 1:10 in molecular grade water), 12.5 μl 2× MyTaq Mix, 0.1 μl primers NF2 and NR2 (5'-AGGGACAGAAGTCGAA GGGG-3' and 5'-GCAGCCAAGCCGGAAACATC-3', both 100 μM) and 11.3 μl molecular grade water. The reaction mixture was heated at 94˚C for 0.5 min, followed by 35 cycles of 94˚C for 0.5 mins, 58˚C for 0.5 min and 72˚C for 0.5 min, with a final extension step of 72˚C for 10 min using a G-Storm thermal cycler. Nested PCR amplification products were resolved by agarose gel electrophoresis through 1% (w/v) UltraPure agarose (Thermofisher Scientific, USA), prepared using 1x TBE (Tris-Borate EDTA) buffer and stained using SafeView (0.01% v/v; NBS Biologicals, UK); visualized using a Syngene U-Genius imaging system. Approximately 10% of amplicons were sequenced to confirm identity (as described below).

## Quantification of *T. gondii* and host genome copy number using qPCR

Quantitative PCR (qPCR) was used to screen samples found to contain *T. gondii* DNA by nested PCR to identify those with parasite genome copy numbers sufficient for further genetic analyses. *Toxoplasma gondii* genome number was quantified using primers targeting the 529 bp gene repeat (S1 Table), standardised using primers targeting the *Gallus gallus domesticus* glyceraldehyde 3-phosphate dehydrogenase (GAPDH), *Sus scrofa* hypoxanthine phosphoribo-syltransferase (HPRT) or *Homo sapiens* p53 tumour suppressing genes for chicken, pig and human samples, respectively (S1 Table). Standard series of ten-fold serial dilutions were prepared using purified PCR amplicons for each assay, as described previously [30]. Quantitative PCR amplification was performed in 96 well plates using the Bio-Rad CFX96 Real time detection system (Bio-Rad laboratories, USA) with SsoFast EvaGreen Supermix (Bio-Rad, USA) in 20 μl reaction volumes per well. Each reaction consisted of 10 μl of EvaGreen Supermix, 0.5 μl each forward and reverse primer (100 μM; S1 Table), 8 μl molecular grade water and l μl template DNA. Cycling conditions for the reactions included initial denaturation at 95˚C for 2 min, followed by 40 cycles of denaturation at 95˚C for 15 sec and annealing/extension at 60˚C for 30 sec. Melt curves were determined at 65–95˚C. Each sample was tested in triplicate, using molecular grade water as the non-template (negative) control. Standard dilution series were run on each plate including $1x10^8$ to $1x10^2$ copies for each host or *T. gondii* assay (as appropriate) using a single well per concentration. Quantification cycle value (Cq) was used to determine the *T. gondii* status of each sample following standardisation against host genome copy number. Following a period of trial and error, samples with more than 0.05 *T. gondii* genomes per host genome (0.1 for chickens), calculated assuming ~100 copies of the 529 bp repeat per *T. gondii* genome [31], were selected for genotyping and amplicon sequencing.

## Genotyping

Monoplex nested PCR assays targeting the *T. gondii* surface antigen 3 (SAG3), SAG2 (5' and 3' fragments), beta-tubulin (BTUB), dense granule protein 6 (GRA6) and apicoplast (Apico) genotyping loci were applied as previously described by Su *et al.* [32]. Reaction mixtures for primary amplification were optimised in 50 μl reaction volumes containing 25 μl 2× MyTaq Mix (Bioline), 0.2 μl forward and reverse primers (100μM; S2 Table), 4 μl extracted DNA and 20.6 μl molecular grade water. Cycle parameters were denaturation at 95˚C for 4 mins, followed by 30 cycles of 94˚C for 0.5 mins, 55˚C for 1 min and 72˚C for 2 mins, using a G-storm thermal cycler. The first round product was diluted 1:1 in molecular grade water. The second round nested PCR was optimised in 50 μl reaction volumes containing 4 μl of diluted first round PCR product, 25 μl 2× MyTaq Mix (Bioline), 0.2 μl forward and reverse primers (100μM) and 20.6 μl molecular grade water. Cycle parameters were denaturation at 95˚C for 4 mins, followed by 35 cycles of 94˚C for 0.5 min, 60˚C for 1 min and 72˚C for 1.5 min, using a G-storm thermal cycler. PCR amplifications were resolved through 1.5% (w/v) agarose gel (except Apico, resolved using 3%) stained with SafeView (0.01%) and visualized using a gel imaging system (Syngene U-Genius). Genomic DNA extracted from the reference *T. gondii* strains GT1, ME49 and VEG, representing genotypes I, II and III respectively, were used as positive controls, while molecular grade water was used as the negative control. Primers for PCR were synthesized by Sigma (S2 Table).

**RFLP typing.** Nested PCR amplicons were first purified and concentrated using a MinElute PCR Purification kit (Qiagen, USA) as described by the manufacturer. Approximately 5 μl of each purified nested PCR product was digested using the relevant restriction enzyme(s) (CutSmart buffer, NEB Labs; enzymes as described elsewhere [33]) in a volume of 20 μl. Digested PCR amplicons were resolved by agarose gel electrophoresis to score each sample/locus genotype [33]. In situations where the RFLP profile was ambiguous, an *in-silico* digest was performed by sequencing the amplicon (GATC Biotech, Konstanz, Germany) followed by manual annotation of the candidate restriction sites, determining the SNP profile and the associated cut/uncut RFLP genotype.

**Gel electrophoresis resolution of PCR-RFLP profiles.** Digested amplicons were resolved by electrophoresis using a 2.5% (w/v) agarose gel (3.0% for Apico), prepared using 1× TBE buffer and stained with 0.01% (v/v) SafeView, followed by visualisation using a Syngene U-Genius imaging system. Fragment sizes were identified by comparison with a low range molecular marker, GeneRuler Low range DNA ladder (ThermoFisher). PCR amplicons produced using genomic DNA extracted from the *T. gondii* reference strains GT1, ME49 and VEG, representing genotypes I, II and III respectively, were included as controls in the digest to confirm profiles for each type.

## Amplicon sequencing, alignment and analysis

Purified PCR amplicons were Sanger sequenced (GATC Biotech, Konstanz, Germany) using the inner nested primers employed in their amplification. Sequence data were curated, assembled and analysed using CLC Main Workbench (v8.0.1), with identity confirmed using BLAST against the National Center for Biotechnology Information (NCBI; https://blast.ncbi.nlm.nih.gov/Blast.cgi) non-redundant nucleotide database.

Sequences generated from PCR-RFLP amplicons were used to confirm PCR-RFLP genotype and subjected to further multi-sequence locus type (MLST) analysis. Reference sequences for the GT1, ME49 and VEG strains were downloaded from ToxoDB (http://toxodb.org/toxo/) as examples of types I, II and III, respectively. All sequences were aligned using CLC Main Workbench (v8.0.1) with the very accurate (slow) settings and default parameters, and then

manually curated. SAG3 and BTUB sequences were aligned with reference sequences to define genetic diversity and for phylogenetic analysis. All sequences downloaded from GenBank are identified by their unique accession number. Single nucleotide polymorphisms (SNPs) and insertion/deletions (indels) were identified. Genetic parameters, including haplotype number and diversity (Hd), and nucleotide diversity ($\pi$) with the Jukes Cantor correction, were calculated using DnaSP (v6.12.03). All sequences generated here have been submitted to the European Nucleotide Archive under the project reference PRJEB42973, with accession numbers HG992860-HG992962.

## Phylogenetic analysis

MEGA-X was used to infer Maximum Likelihood (ML) and Neighbor Joining (NJ) phylogenies for SAG3 and BTUB. Using default parameters, the Hasegawa-Kishino-Yano (HKY) model with gamma distribution (G) and evolutionary invariable (I) was identified as the best DNA model for SAG3, while the Jukes and Cantor (JC) model was identified for BTUB, based upon the Bayesian information criterion (BIC). Trees were created with 1,000 bootstrap iterations. Bayesian phylogenetic analysis (MrBayes) was determined using TOPALi v2.5 [34]. Model selection confirmed the HKY+G+I and JC models for SAG3 and BTUB, respectively. Analysis used 2 runs with 5,000,000 generations and 35% burnin for construction of MrBayes trees.

## Network analysis

Samples sequenced at all five PCR-RFLP loci were used for Network analysis. Sequences were aligned using CLC Main Workbench (v8.0.1) for each locus, concatenated and exported in Phylip format. The concatenated alignment was imported into Network (v10.2), where a Network was drawn using the Median Joining (MJ) technique with default parameters. The concatenated nucleotide alignment used is available from the Mendeley data repository under the accession http://dx.doi.org/10.17632/fscy8463cd.2.

## Results

### Detection of *Toxoplasma gondii* genomic DNA in samples collected from pigs, free range chickens and seropositive pregnant women in Benue state, Nigeria

*Toxoplasma gondii* DNA was detected in all three of the populations sampled. Sampling thigh muscle during preparation of meat for human consumption found 30.6% of 173 free range chickens and 31.3% of 211 pigs contained *T. gondii* genomic DNA at levels detectable using nested PCR (Table 1). For seropositive pregnant women, 25.3% of 91 blood samples were positive for detectable levels of *T. gondii* genomic DNA. Excluding samples with inconclusive results for serology, 3/14 samples collected from IgM seropositive women were also positive

**Table 1. Detection of *Toxoplasma gondii* genomic DNA in thigh muscle samples collected from free-range chickens and pigs, and in blood collected from seropositive pregnant women.**

| Host | No. sampled | No. positive by nPCR (%) | *T. gondii* genomes per host genome: mean (max, min) | No. above qPCR threshold* |
|---|---|---|---|---|
| Chicken | 173 | 53 (30.6) | 0.085 (0.523, 0.001) | 25 |
| Pig | 211 | 66 (31.3) | 0.038 (0.224, 0.006) | 14 |
| Human | 91 | 23 (25.3) | 0.046 (0.239, 0.007) | 12 |
| Total | 475 | 142 (29.9) | - | 51 |

*Threshold for chickens: 0.1 *T. gondii* genomes per host genome; for pigs and humans: 0.05 *T. gondii* genomes per host genome.

**Table 2. Summary of *Toxoplasma gondii* detection by nested PCR (nPCR) in blood collected from pregnant women subdivided IgM and IgG *anti-Toxoplasma* serology.**

| Assay | No. sampled | Definitive serology result* | Ig Positive (nPCR positive) | Ig Negative (nPCR positive) |
|---|---|---|---|---|
| IgM | 91 | 83 | 14 (3) | 69 (18) |
| IgG | 91 | 89 | 73 (18) | 16 (4) |

*Serological results for the remaining samples were inconclusive.

for *T. gondii* by PCR (21%, Table 2). In total 18/73 samples collected from IgG reactive women were also *T. gondii* positive by PCR (25%). Five women were found to be IgM/IgG positive, all of whom were negative by PCR. No association was detected between stage of pregnancy (trimester) and detection of *T. gondii* genomic DNA by nested PCR (Table 3). Quantitative PCR targeting the 529 bp repeat, assuming ~100 repeat copies per genome, suggested between 0.52 and 0.001 parasite genomes per host genome in each nPCR positive sample (Table 1). Samples exceeding 0.05 parasite genomes per host genome (0.1 for chickens) progressed to PCR-RFLP.

## PCR-RFLP genotyping

A panel of 51 genomic DNA samples were subjected to preliminary PCR-RFLP at the SAG3 locus. In total 32 samples were typed, all of which were identified as type II (Table 4). All 32 SAG3-typed samples were subsequently genotyped using the BTUB and 5' SAG2 PCR-RFLPs, resulting in 17 (BTUB, all type II) and 16 (5' SAG2, all type I or II) genotypes, respectively (Table 4). Finally, samples genotyped for at least two loci were further characterised using the 3' SAG2, GRA6 and Apico PCR-RFLPs. Combined, these results indicated that *T. gondii* circulating in the sampled populations primarily presented with a type II genetic background, although all included a hybrid type I/II or II/III PCR-RFLP haplotype (Table 4). Only the SAG2, GRA6 and Apico markers were discriminatory between the three classical *T. gondii* genetic types. Comparison with TgCkNg1, the only previously published *T. gondii* RFLP type from Nigeria (SAG3, BTUB and GRA6 type III, 5'+3' SAG2 and Apico type I; [20]), revealed comparable types at a minority of SAG2 and Apico markers only.

## Multi-locus sequence typing

Sequencing SAG3 PCR-RFLP amplicons revealed five distinct genotypes, all within the type II RFLP group (Tables 4 and 5). The sequences are available under the accession numbers HG992860-HG992891. Comparison with other African TgSAG3 sequences available from GenBank, representing Ethiopia, Gabon and Tunisia, revealed that the dominant Nigerian genotype shared a common identity with the reference ME49 strain and the most common sequence recovered previously in Ethiopia (Fig 1, shown as cluster A). Four additional new sequence genotypes were identified, three of which were represented by two or more

**Table 3. Detection of *Toxoplasma gondii* genomic DNA in blood collected from seropositive pregnant women per trimester of pregnancy.**

| Trimester | No. sampled | nPCR positive (%) | nPCR negative (%) |
|---|---|---|---|
| 1 | 7 | 2 (29) | 5 (71) |
| 2 | 25 | 5 (20) | 20 (80) |
| 3 | 53 | 16 (30) | 37 (70) |
| Not known | 6 | 0 (0) | 6 (100) |
| Total | 91 | 23 | 68 |

**Table 4. Summary of *Toxoplasma gondii* PCR-RFLP genotyping results from free-range chickens, pigs and seropositive pregnant women sampled in Benue State, Nigeria.**

| Host | ID | PCR-RFLP locus type | | | | | | |
|---|---|---|---|---|---|---|---|---|
| | | SAG3 | BTUB | 5'-SAG2 | 3'-SAG2 | 5'+3' SAG2 | GRA6 | Apico |
| Chicken | CGBK6 | II | - | - | | | | |
| | CKA18 | II | II | I or II | I or III | I | I | I |
| | CKA2 | II | - | - | | | | |
| | CMKD16 | II | II | I or II | II | II | II | III |
| | CMKD17 | II | II | - | | | | |
| | COKPK16 | II | II | I or II | II | II | II | III |
| | COKPK23 | II | II | I or II | I or III | I | II | I |
| | COKPK24 | II | II | I or II | I or III | I | - | |
| | COKPK28 | II | II | - | | | | |
| | COTU29 | II | - | I or II | - | - | | |
| Human | FMC16 | II | - | - | | | | |
| | FMC17 | II | - | - | | | | |
| | FMC21 | II | - | - | | | | |
| | FMC27 | II | II | I or II | I or III | I | I | I |
| | FMC32 | II | II | I or II | II | II | II | I |
| | FMC46 | II | II | I or II | I or III | I | I | I |
| | GHM14 | II | II | I or II | II | II | II | III |
| | GHM35 | II | II | I or II | II | II | II | III |
| | GHO55 | II | - | - | | | | |
| | GHU18 | II | - | I or II | - | - | | |
| Pig | PGBK18 | II | - | - | | | | |
| | PGBK26 | II | - | - | | | | |
| | PGBK3 | II | - | - | | | | |
| | PGE25 | II | - | - | | | | |
| | PGE48 | II | - | - | | | | |
| | PKON16 | II | II | I or II | II | II | II | I |
| | PKON22 | II | II | - | | | | |
| | PKON37 | II | II | I or II | II | II | II | I |
| | PMDK21 | II | II | I or II | II | II | - | |
| | PMDK29 | II | II | I or II | II | II | II | III |
| | PZB27 | II | - | - | | | | |
| | PZB58 | II | - | - | | | | |
| Total | | 32 | 17 | 16 | 14 | 14 | 12 | 12 |

- = tested, no PCR-RFLP result.

independent samples. SAG3 sequences described from Gabon and Tunisia were found to be distinct, with the majority from types I or III (Fig 1). Comparison of SAG3 genotypes by host found no obvious host species-specific sub-structures.

Sequencing the BTUB PCR-RFLP also identified five distinct genotypes (Table 5, accession numbers HG992892-HG992908). As for SAG3, all BTUB sequences mapped to the RFLP type II group (Table 4). However, comparison with the SAG3 genotypes revealed a different hierarchy of relatedness (Figs 1 and 2). For example, samples defined by the SAG3 cluster A sequence genotype were found in all five BTUB genotype groups (Fig 2). Comparison of BTUB genotypes by host also found no obvious host species-specific sub-structures.

**Table 5. Summary of multi-locus sequence type (MLST) analysis including five PCR-RFLP loci individually and after concatenation.**

| Locus | N | n | S | H | Hd | k | π (JC) | Variant sites |
|---|---|---|---|---|---|---|---|---|
| SAG3 | 32 | 226 | 7 (4) | 5 | 0.48 ± (0.10) | 0.835 | 0.004 | 83, 97, 122, 145, 173, 487, 199 |
| BTUB | 17 | 411 | 3 (3) | 5 | 0.83 ± (0.04) | 1.309 | 0.003 | 160, 176, 179 |
| 5' SAG2 | 16 | 242 | 0 (0) | 1 | 0.00 ± (0.00) | 0.000 | 0.000 | - |
| 3' SAG2 | 14 | 222 | 3 (3) | 2 | 0.49 ± (0.09) | 1.484 | 0.007 | 4, 148, 168 |
| GRA6 | 12 | 344 | 5 (4) | 3 | 0.53 ± (0.14) | 1.803 | 0.005 | 161, 253, 261, 277, 316 |
| Apico | 12 | 845 | 1 (1) | 2 | 0.53 ± (0.08) | 0.530 | 0.001 | 191 |
| Concatenation | 12 | 2290 | 16 (12) | 9 | 0.95 ± (0.05) | 6.015 | 0.003 | na |

N = number of sequences included. n = number of nucleotides aligned. S = number of variant sites, with the number of parsimony informative variant sites shown in parentheses. H = the number of genotypes/haplotypes. Hd = the genotype/haplotype diversity with the standard deviation shown in parentheses. k = the average number of pairwise differences. π (JC) = nucleotide diversity, calculated with the Jukes Cantor correction. na = not applicable.

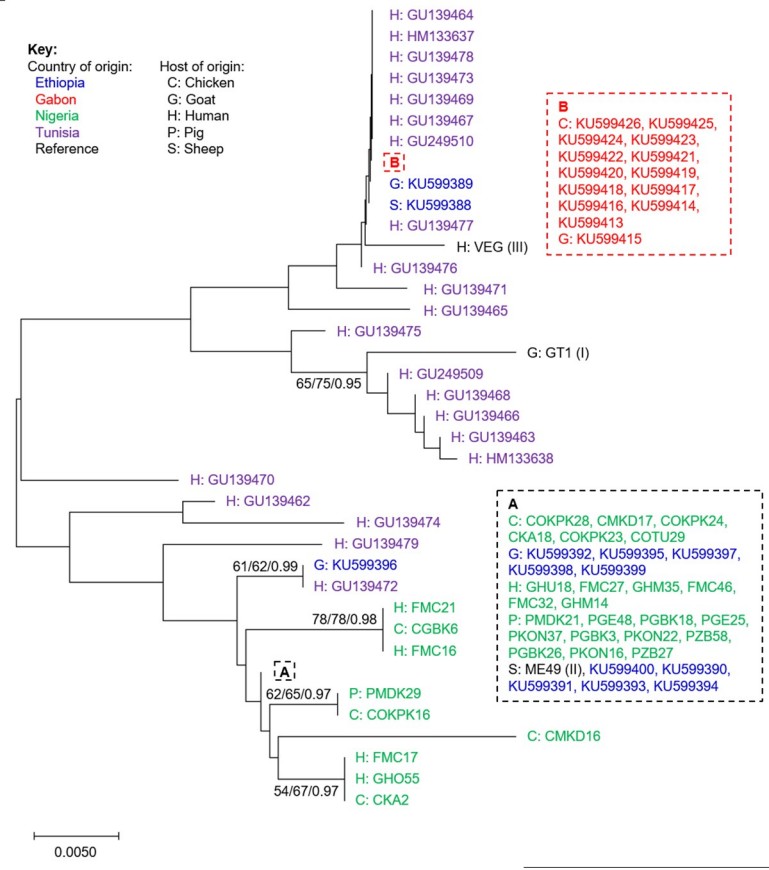

**Fig 1. Optimal Maximum Likelihood (ML) phylogenetic tree for *Toxoplasma gondii* PCR-RFLP SAG3 sequences from Nigerian chickens, pigs and seropositive pregnant women.** Generated using an alignment of 226 bp from 84 sequences in total, including 32 new sequences from Nigeria plus published sequences from Ethiopia, Gabon and Tunisia (downloaded from GenBank for comparison, as indicated by accession number). Sequences from the reference GT1, ME49 and VEG strains, representing types I, II and III, were accessed from ToxoDB. Figures at nodes in the format a/b/c represent a: the percentage of trees in which the associated taxa clustered together using ML, b: using Neighbor Joining, and c: posterior probability by Bayesian inference. A and B denote clusters of identical sequences, as indicated in the associated boxes.

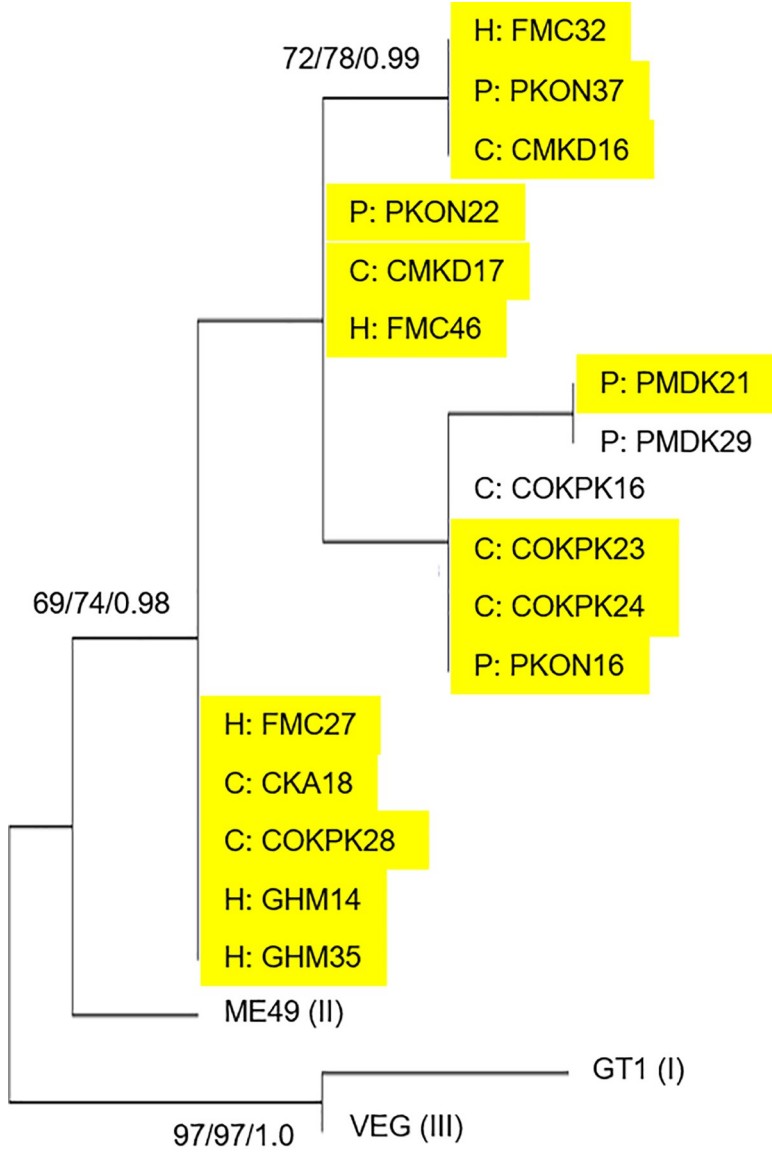

**Fig 2. Optimal Maximum Likelihood (ML) phylogenetic tree for *Toxoplasma gondii* PCR-RFLP BTUB sequences from Nigerian chickens, pigs and seropositive pregnant women.** Generated using an alignment of 411 bp from 20 sequences in total, including 17 new sequences from Nigeria. Sequences from the reference GT1, ME49 and VEG strains, representing types I, II and III, were accessed from ToxoDB. Figures at nodes in the format a/b/c represent a: the percentage of trees in which the associated taxa clustered together using ML, b: using Neighbor Joining, and c: posterior probability by Bayesian inference. Sequences highlighted in yellow represent samples with a conserved SAG3 sequence as shown in Fig 1, cluster A.

Limited genetic diversity was detected within the 5' and 3' SAG2, GRA6 and Apico PCR-RFLP amplicons, mostly restricted to polymorphisms already described and associated with RFLP type (Table 5, accession numbers HG992909-HG992962).

A panel of 12 samples were sequenced at all five PCR-RFLP loci, representing genomic DNA extracted from four free-range chickens, three pigs, and five seropositive pregnant women

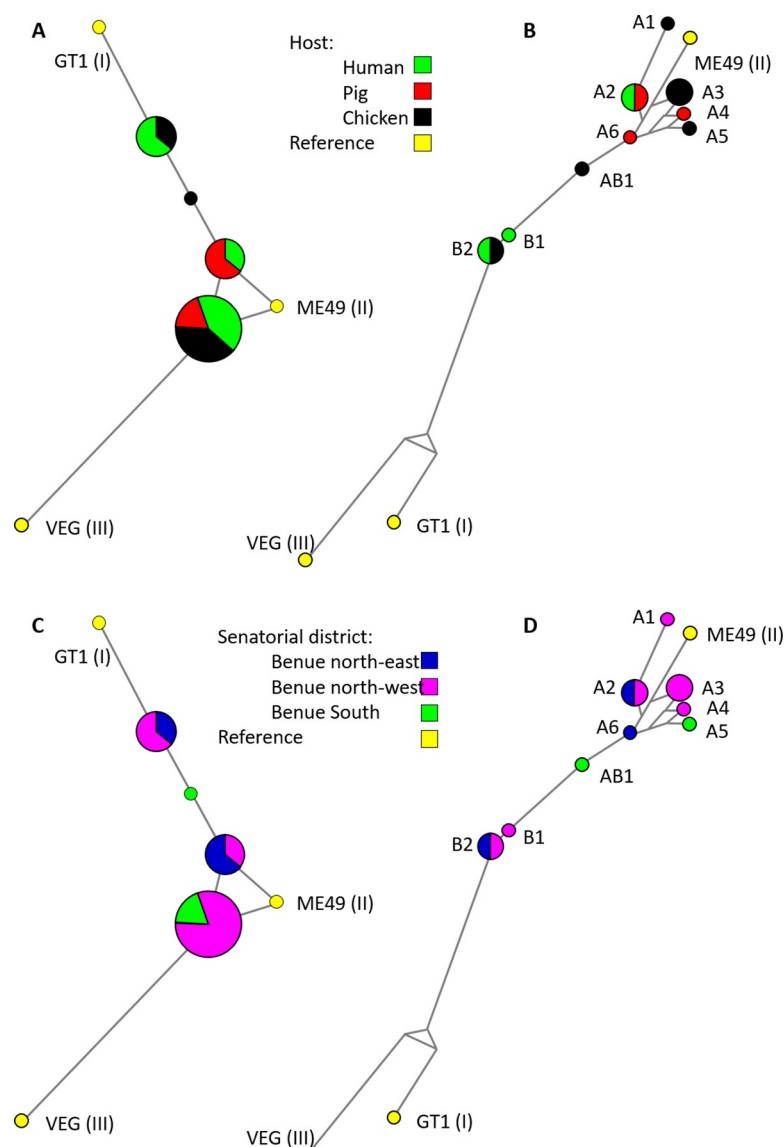

**Fig 3. Haplotype networks describing *Toxoplasma gondii* sampled from chickens, pigs and humans in Nigeria.** A. Haplotypes constructed using five RFLP markers (SAG3, BTUB, 5'+3' SAG2, GRA6 and Apico), differentiated by host of origin. B. Haplotypes constructed using a concatenated alignment of 2290 bp representing the sequenced SAG3, BTUB, 5' SAG2, 3' SAG2, GRA6 and Apico PCR-RFLP amplicons, differentiated by host of origin. C. Haplotypes constructed using five RFLP markers, differentiated by senatorial zone of origin. D. Haplotypes constructed using a concatenated alignment of 2290 bp, differentiated by senatorial zone of origin. Haplotypes detected in Nigerian samples annotated in clusters A, B and intermediate AB.

(Table 4). Consideration of PCR-RFLP type indicated four haplotypes, including between one and five samples per haplogroup (Fig 3A). Nine samples were most closely related to type II, while three were closer to type I. Consideration of a concatenated sequence alignment for all 12 samples, including all five loci, provided greater specificity and revealed nine haplotypes (Table 5 and Fig 3B). The six haplotypes most closely related to reference ME49 type II have been annotated A1-6 here. Two haplotypes in a second cluster have been annotated B1-2, with one intermediate haplotype (AB1; Fig 3B). The concatenated nucleotide alignment used is available from the Mendeley data repository under the accession http://dx.doi.org/10.17632/fscy8463cd.2. The majority of the

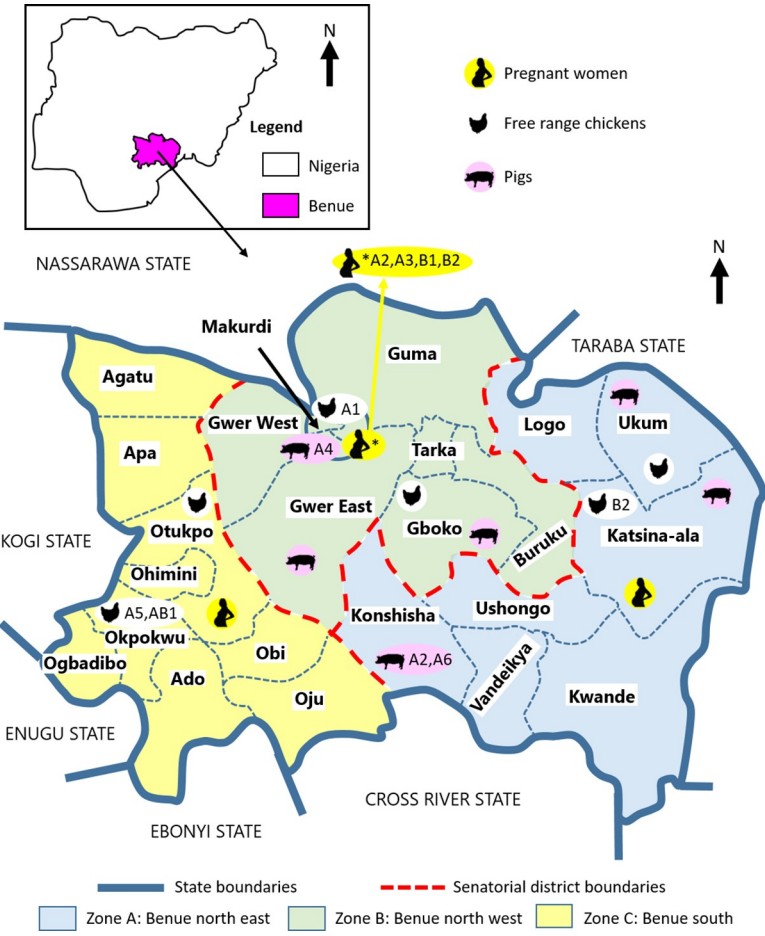

**Fig 4. Sample sites and *Toxoplasma gondii* haplotype occurrence within Local Government Areas (LGAs) in Benue state, Nigeria.** LGAs sampled are shown, grouped into Zones A (Katsina-Ala, Konshisha, Ukum), B (Gboko, Gwer East, Markurdi) and C (Okpokwu, Oturkpo). The distribution of haplotypes A1-6, B1-2 and AB1 annotated in Fig 3. is indicated. (Map redrawn here based upon resources accessed from grid3.gov.ng).

sequence-based haplotypes were most closely related to the type II reference ME49 (A1-6), although none were identical. Six of the nine haplotypes were represented by single samples. The remaining three haplotypes were represented by two samples each. Comparison of sample meta-data for host and senatorial district of origin revealed the absence of detectable sub-populations in RFLP and MLST datasets (Fig 3A–3D), confirmed by mapping haplotypes to LGAs within Benue state (Fig 4).

## Discussion

Information defining the occurrence and risk posed by *T. gondii* in Africa is scarce, with studies of genetic diversity even less common. *Toxoplasma* infection during pregnancy is associated with serious consequences in humans, including abortion, still birth and a range of congenital abnormalities [10,35]. Chickens and pigs are important sources of infection for humans and, although infection in these species is commonly asymptomatic, clinical disease in sows and piglets can be severe [36,1]. We have previously found high seropositivity for *T. gondii* in the pregnant women tested here, primarily represented by chronic infections (i.e. IgM negative, IgG positive; [27]). Using a sensitive nested PCR targeting the *T. gondii* 529 bp

multi-copy repeat as an indirect measure of occurrence [37,38,29], recognising that the presence of DNA does not necessarily indicate active infection, we suggest a high occurrence of blood parasitaemia in pregnant women in Benue state, Nigeria. Previous studies using human blood samples as template for PCR detection of *T. gondii* have reported that 48.6% of IgM seropositive patients were also positive by PCR, falling to 3.6% in IgM negative, IgG positive chronically infected patients [39]. In a separate study, 31% of asymptomatic chronically infected people tested positive by PCR [40]. Here, similar levels of detection were reported from IgM and IgG positve women. The stage of pregnancy (trimester) was not associated with variation in detection of *T. gondii* DNA. Such high occurrence emphasises the importance of public health policies focused on *T. gondii*. The occurrence of *T. gondii* detected in chickens and pigs was also high, and might have been higher if tissues such as brain, lungs or heart had been sampled [41,12]. Here, we sampled chicken and pig muscle (meat) intended for human consumption as a direct indication of occurrence and risk to human health. The present study indicated a higher occurrence than that reported in Iran [42,43], comparable with that from Eygpt [44] and lower than reported from Saudi Arabia [45].

Few *T. gondii* genotypes have been described from Africa, prompting Galal and colleagues to describe the continent as a missing link in population structure for the parasite [46]. For Nigeria, just one RFLP genotype had been published prior to this study, defining a sample collected in 2005 from free-range chickens in Vom, Plateau state, with an atypical mixed type III/I haplotype [20]. This haplotype has not been detected in the present study. Wider genetic analysis of *T. gondii* collected in Africa has revealed three nonarchetypal genotypes, termed Africa 1, 2 and 3 [21,22]. These new African genotypes have been defined using microsatellite markers, not PCR-RFLP, so it is difficult to compare directly with the haplotypes reported here. Comparative analysis including reference strains of the type I, II and III genotypes tentatively suggest these African genotypes are most closely related to type I [21,22]. Here, *T. gondii* haplotypes generated from samples collected during concurrent studies with pregnant women, chickens and pigs presented with a predominantly type II genetic background, in agreement with several previous studies from Africa [20,46]. Type II *T. gondii* has previously been detected in Uganda [47], Tunisia [21]), Egypt [48], Ethiopia [49] and Ghana [50], but not in Gabon [22]. Although *T. gondii* type II strains have often been considered avirulent, they have been reported in both benign and severe congenital toxoplasmosis in France [51]. Analysis using the SAG3 locus including published sequences from Ethiopia, Gabon and Tunisia provide further support for the conclusion that *T. gondii* strains circulating in Nigeria differ from the African genotypes 1, 2 and 3 (Fig 1). Four distinct RFLP haplotypes are reported here, each represented by between one and five samples. Comparison using the ToxoDB RFLP database (https://toxodb.org/toxo/app/search/rflp-isolate, accessed 31/01/2021) indicated one previous example of the most common haplotype reported here: strain B73, collected from a bear in the USA (reference EUSMPL0040-1-320) [52]. Genotyping these samples at additional loci would have permitted further assessment of their relatedness with strains in the published literature. A second haplotype, detected in three samples, was found to be relatively common in the published literature, appearing in 321 of 1,496 RFLP records representing multiple hosts and regions. Again, further genotyping would have permitted greater discrimination. The two remaining haplotypes were new, not represented in the ToxoDB database.

Production of clear RFLP profiles that were readily visible on an agarose gel was a major challenge, likely due to low parasitaemia in many samples. *Toxoplasma gondii* DNA was extracted directly from the sampled populations, representing natural infections. This challenge has been common in *Toxoplasma* research [53,9]. Bioassay using mice could have increased the number of samples suitable for genetic analysis, but was not within the scope of the project. As an alternative, direct PCR amplicon sequencing was adopted, using *in-silico*

digestion to define RFLP type. Quantifying parasite load in each sample using qPCR became a necessary screening step after successive failures of PCR-RFLP to produce clear and distinct profiles, despite several attempts at optimization of the PCR and RFLP steps. After prioritising those samples with the highest *T. gondii* genome content just 12 samples could be genotyped at five independent loci.

Amplicons sequenced for PCR-RFLP were also used for MLST analysis, starting with the SAG3 locus. Haplotype diversity was greatest for SAG3 and BTUB, although sample numbers were also highest. Nucleotide diversity was highest at the GRA6 locus, while no polymorphism was detected at the 5' SAG2 locus. MLST analyses were more discriminatory than RFLP. Comparison of SAG3 sequences with published resources from Africa available in GenBank revealed one dominant sequence type in Nigeria, shared with the reference ME49 strain and multiple sequences generated previously from Ethiopia. Sequences from Gabon and Tunisia were found to be distinct. The lack of similarity between sequences from Nigeria and Gabon was striking, given their geographical proximity, although it is worth noting the lack of SAG3 sequence variation detected in Gabon (Fig 1; [54]), possibly representing expansion of a specific sequence type. Considering sequences from many of the same Nigerian samples at a second locus, BTUB, disrupted the conserved SAG3 sequence group and suggested greater haplotype diversity, although nucleotide diversity remained low outside of the RFLP restriction sites. This is in contrast to findings from a study conducted in Ghana, where *T. gondii* circulating in both animal and human populations were genetically diverse [50]. Consideration of haplotypes created by PCR-RFLP and MLST approaches revealed no clear trends based upon host or senatorial zone of origin (Fig 3), suggesting a single or overlapping *T. gondii* population(s) within Benue state. Evidence of shared haplotypes in samples collected from pregnant women as well as chicken and pig meat samples reinforces the well-known zoonotic link between animal and human infections. While the sample set was small, chickens were found to host the greatest MLST haplotype diversity, supporting their value as sentinels [4]. Future public health activities should continue to focus on best practices around meat processing and management in the home and food establishment settings.

The present study has produced the first genotypes for *T. gondii* circulating in free-range chickens, pigs and pregnant women in Benue State, Nigeria, suggesting closest links to archetypal type II *T. gondii* strains. The study indicated that free-range chickens and/or pigs shared *T. gondii* genotypes with pregnant women, representing potential sources of infection to humans in the study area. Direct source attribution cannot be confirmed given that the animals sampled may have been imported from other LGAs, and that the pregnant women sampled are likely to have travelled to other regions in the past, but it is likely that these genotypes were circulating in the environment. Africa remains under-explored for *T. gondii* genetic diversity and this study provides the first detailed definition of genotypes circulating in human, chicken and pig populations in Nigeria.

## Supporting information

**S1 Table. Primer sequences and cycling conditions for host and *Toxoplasma gondii* specific qPCR.**
(DOCX)

**S2 Table. PCR primers and restriction enzymes used for nested multi-locus PCR-RFLP and *in-silico* digest.**
(DOCX)

## Acknowledgments

The authors would like to thank Clare Hamilton for providing DNA samples for use as positive controls and Hassan Muazu for assistance with Arc Gis. The Royal Veterinary College has assigned this manuscript the reference number 1443835.

## Author Contributions

**Conceptualization:** Ifeoma N. Nzelu, Jacob K. P. Kwaga, Junaidu Kabir, Idris A. Lawal, Damer P. Blake.

**Data curation:** Ifeoma N. Nzelu, Damer P. Blake.

**Formal analysis:** Ifeoma N. Nzelu, Damer P. Blake.

**Funding acquisition:** Ifeoma N. Nzelu.

**Investigation:** Ifeoma N. Nzelu, Christy Beazley, Laura Evans.

**Methodology:** Ifeoma N. Nzelu, Christy Beazley, Laura Evans, Damer P. Blake.

**Project administration:** Ifeoma N. Nzelu.

**Resources:** Damer P. Blake.

**Supervision:** Damer P. Blake.

**Validation:** Ifeoma N. Nzelu, Damer P. Blake.

**Writing – original draft:** Ifeoma N. Nzelu, Damer P. Blake.

**Writing – review & editing:** Ifeoma N. Nzelu, Jacob K. P. Kwaga, Junaidu Kabir, Idris A. Lawal, Damer P. Blake.

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
