## [Decision Letter · Decision Letter 0]

13 Apr 2021

Dear Professor Blake,

Thank you very much for submitting your manuscript "Prevalence and genetic characterisation of Toxoplasma gondii circulating in free-range chickens, pigs and pregnant women in Benue state, Nigeria" for consideration at PLOS Neglected Tropical Diseases. As with all papers reviewed by the journal, your manuscript was reviewed by members of the editorial board and by several independent reviewers. The reviewers appreciated the attention to an important topic. Based on the reviews, we are likely to accept this manuscript for publication, providing that you modify the manuscript according to the review recommendations. 

Sincerely,

Arnau Casanovas-Massana, PhD

Associate Editor

Pikka Jokelainen, PhD

Deputy Editor

Reviewer's Responses to Questions

**Key Review Criteria Required for Acceptance?**

**Methods**

-Are the objectives of the study clearly articulated with a clear testable hypothesis stated?

-Is the study design appropriate to address the stated objectives?

-Is the population clearly described and appropriate for the hypothesis being tested?

-Is the sample size sufficient to ensure adequate power to address the hypothesis being tested?

-Were correct statistical analysis used to support conclusions?

-Are there concerns about ethical or regulatory requirements being met?

Reviewer #1: yes

Reviewer #2: The methods are adequate to answer questions

Reviewer #3: The present study is clearly articulated with a clear testable hypothesis, and design appropriate to address the stated objectives. The population clearly described and appropriate for the hypothesis being tested. The sample size is proper and sufficient to ensure adequate power to address the hypothesis being tested. The data analysis used a support author's conclusions, and no concerns about ethical or regulatory requirements were met. However, authors mention that it was collected sera samples from pregnant women, the T. gondii human serology results were not shown. This result is not essential for the stated objectives, but certainly could improve the understanding of PCR-based methods and sample strategy collection performance to detect parasite DNA in human blood samples.

**Results**

-Does the analysis presented match the analysis plan?

-Are the results clearly and completely presented?

-Are the figures (Tables, Images) of sufficient quality for clarity?

Reviewer #1: yes

Reviewer #2: The analysis correspond with the aims and methods

Reviewer #3: The analysis presented matches the analysis plan the figures and tables are clearly presented and with quality. I believe that the inclusion of one more figure describing the sampled LGAs in correlation with the corresponding genotypes isolated from chickens, pigs and pregnant women could contribute to improve the data interpretation concerning the zoonotic character of T. gondii infections.

**Conclusions**

-Are the conclusions supported by the data presented?

-Are the limitations of analysis clearly described?

-Do the authors discuss how these data can be helpful to advance our understanding of the topic under study?

-Is public health relevance addressed?

Reviewer #1: yes

Reviewer #2: The conclusions are derived from the results

Reviewer #3: The conclusions are supported by the data presented, the public health relevance is stated. Because of the scarcity of genotyping data in Africa, particularly in Nigeria and considering the present study, it is clear that more detailed studies are necessary to better understand the genetic population structure and the dynamics of parasite circulation in animals and humans in Nigeria and in Africa in general.

**Editorial and Data Presentation Modifications?**

Reviewer #1: yes

Reviewer #2: Please see at summary nd general cmments

Reviewer #3: The authors report a set of data that are relevant and well presented in the manuscript.

However I suggest some minor edits of existing data which enhance clarity. They are listed below

1- in the title where it reads Prevalence and genetic characterisation of Toxoplasma gondii circulating in free-range chickens, pigs and pregnant women in Benue state, Nigeria

I would suggest that the word "prevalence" should not be employed in the title (and also in the manuscript), because the term "prevalence" has an epidemiological connotation, and, what has been done cannot represent a prevalence in epidemiologic terms. I suggest using "occurrence" instead of "prevalence", then it would be: Occurence and genetic characterisation of Toxoplasma gondii circulating in free-range chickens, pigs and pregnant women in Benue state, Nigeria

2-Authors reports that bioassays remain the most sensitive means of detecting Toxoplasma cysts in animal tissues. However it is even more evident that molecular methodologies should be employed for surveys in animal meat samples as demonstrated by Schares and colleagues: International Journal for Parasitology 48 (2018) 751-762, where it is presented a paper entitled: Toxoplasma gondii infections in chickens – performance of various antibody detection techniques in serum and meat juice relative to bioassay and DNA detection methods. In my opinion authors could refer to Schares publication to better value their findings. However authors have not measured antibodies in animals for the present study, they have several previous serological studies in Nigeria showing that the seroprevalence is on average in the range of 30% for animals (chicken and pigs) and similar to humans. Only one study in Nigeria demonstrated T. gondii higher human seroprevalence in slaughterhouse workers (55%), probably revealing a higher risk for T. gondii exposure and infection for this specific group. Then, based on this premissa authors´ rate of T. gondii DNA detection in the present study may be very high.

3- Please in the Methods section inform how long, and under what condition (temperature, humidity) Paper Card with human blood samples were kept until processed for DNA extraction. Another important information is: for each patient what was the total quantity of blood in the Paper Card processed for DNA extraction?

4- I suggest the inclusion of a new figure describing the samples collected from the LGAs (Ukum, Kastina-Ala, Makurdi, Gboko, Otukpo and Okpokwu for free-range chickens; Ukum, Kastina-Ala, Konshisha, Makurdi, Gboko and Gwer- East for pigs, and Kasitina-Ala, Makurdi and Otukpo for pregnant women) in correlation with the corresponding genotypes isolated from chickens pig and pregnant women from those LGAs. I believe it could contribute to improve the data interpretation concerning the zoonotic character of T. gondii infections. A figure with the Nigeria map,could spatially gather information from the sampled LGAs with the corresponding genotypes isolated discriminating from chickens, pigs and pregnant women for each LGA. It is, at certain measures, shown in figure 3 but, it is not so easily understandable in that figure. It is noteworthy that the way the study was conducted, this correlation cannot be straightly established, because one cannot exclude that meat of the animals evaluated in each region, might have been originary from another places and sold in the local market. In addition, patients also cirlulate in different regions during their lives. This is an intrinsic limitation of this type of study and there is no way to avoid it. Nonetheless, this does not invalidate the study presented, however, it should be stated in the manuscript clearly as an intrinsic limitation for this type of study.

5-However authors mention that it was collected sera samples from pregnant women, the T. gondii human serology results were not shown. This result is not essential for the stated objectives but certainly could improve the understanding of PCR-based methods and sample strategy collection performance to detect parasite DNA in human blood samples. In this case a new column with serological human information could be added to table 1. Also in this, or in a separate table the stage of pregnancy (gestational age of the fetus) if known could be informed. This information can also be valuable for specific education campaigns for the local primary prevention of congenital toxoplasmosis.

6- It is important to be informed if serological evaluation soon after the blood collection (in 2016) and treatment were possible to be provided to pregnant women and their babies. These informations are important to be reported for establishing critic view for the necessity of public health consistent policies to be adopted in all over the world for prevention of this neglected disease.

7-Authors conclude (lines 713 -715) that: "free-range chickens and/or pigs shared T. gondii genotypes with pregnant women, representing probable sources of infection to humans in the study area." I´d suggest that authors could consider that instead of writing "representing probable sources of infection to humans" it could be: "representing potential sources of infection to humans", also it should be considered and stated that those genotypes are in environment and that, as well as the animals, humans could be infected with those genotypes from environment, by drinking water and or eating soil and vegetables contaminated.

**Summary and General Comments**

Reviewer #1: 1. Line 253—parasitaemia will be appropriate only for blood samples from women but not meat samples from chickens and pigs

2. Perhaps the authors are unaware of 2 recent reviews for toxoplasmosis in chickens ( Parasitology147, 1263-1289, 2020) and pigs (Vet. Parasit. 288, 109185)

Reviewer #2: This is an important contribution for the best knowledge of the Toxoplasma diversity and circulation by adding novel information about genotypes in meat for human consumption and in humans in Nigeria. The article is well written, and methods are adequate, and the authors do a good interpretation of the results. I only have as suggestions: 

- A best explanation how PCR contamination was controlled at the lab and what controls were used for this

-To include some references that support the findings of PCR positive in blood during prevalence studies and to compare with the finding of the authors: 

1. Parasitol. Res. 101, 619–625. http://dx.doi.org/10.1007/s00436-007-0524-9.

2. Acta Tropica 2018; 184:83-87. doi: 10.1016/j.actatropica.2018.01.013.

Reviewer #3: The present study brings out relevant information on the presence of T. gondii genotypes in Nigeria, where there is practically no information about in this field. The data contextualize its importance for public health (congenital toxoplasmosis). The methodology used to detect genetic material of the parasites is described in the literature and used successfully in the manuscript. Authors present an interesting strategy based on collecting and extracting DNA for forensic research, that is: transfer blood in EDTa to Whatman FTA card for subsequent DNA extraction. This strategy proved to be viable and perhaps could be used as a strategy for locals where the preservation of genetic material might be at risk due to the physical distance and or time between collection and extraction processing.

 Thepresented data reinforce the notion that the population structure of T. gondii in Africa, particularly in Nigeria, seems to exhibit lower genetic diversity when compared to the greater diversity observed for south america, and, in the present case, suggesting closest links to archetypal type II T. gondii strains.

PLOS authors have the option to publish the peer review history of their article (what does this mean?). If published, this will include your full peer review and any attached files.

Reviewer #1: No

Reviewer #2: Yes: Jorge Gomez-Marin

Reviewer #3: No

Figure Files:

Data Requirements:

Reproducibility:

References

---

## [Editor Report · Decision Letter 1]

10 May 2021

Dear Professor Blake,

We are pleased to inform you that your manuscript 'Detection and genetic characterisation of Toxoplasma gondii circulating in free-range chickens, pigs and seropositive pregnant women in Benue state, Nigeria' has been provisionally accepted for publication in PLOS Neglected Tropical Diseases.

Best regards,

Arnau Casanovas-Massana, PhD

Associate Editor

Pikka Jokelainen, PhD

Deputy Editor

---

## [Editor Report · Acceptance letter]

28 May 2021

Dear Professor Blake,

We are delighted to inform you that your manuscript, "Detection and genetic characterisation of *Toxoplasma gondii* circulating in free-range chickens, pigs and seropositive pregnant women in Benue state, Nigeria," has been formally accepted for publication in PLOS Neglected Tropical Diseases.

Best regards,

Shaden Kamhawi

co-Editor-in-Chief

Paul Brindley

co-Editor-in-Chief
